# Investigation on Inertial Sorter Coupled with Magnetophoretic Effect for Nonmagnetic Microparticles

**DOI:** 10.3390/mi11060566

**Published:** 2020-05-31

**Authors:** Jiayou Du, Long Li, Qiuyi Zhuo, Ruijin Wang, Zefei Zhu

**Affiliations:** School of Mechanical Engineering, Hangzhou Dianzi University, Hangzhou 310018, China; abc@hdu.edu.cn (J.D.); lilong@hdu.edu.cn (L.L.); zqy66@hdu.edu.cn (Q.Z.); zzf_3691@163.com (Z.Z.)

**Keywords:** magnetophoretic effect, inertial effect, sorting efficiency, nonmagnetic microparticles

## Abstract

The sizes of most prokaryotic cells are several microns. It is very difficult to separate cells with similar sizes. A sorter with a contraction–expansion microchannel and applied magnetic field is designed to sort microparticles with diameters of 3, 4 and 5 microns. To evaluate the sorting efficiency of the designed sorter, numerical simulations for calculating the distributions of microparticles with similar sizes were carried out for various magnetic fields, inlet velocities, sheath flow ratios and structural parameters. The numerical results indicate that micro-particles with diameters of 3, 4 and 5 microns can be sorted efficiently in such a sorter within appropriate parameters. Furthermore, it is shown that a bigger particle size and more powerful magnetic field can result in a greater lateral migration of microparticles. The sorting efficiency of microparticles promotes a lower inlet velocity and greater sheath flow ratios. A smaller contraction–expansion ratio can induce a greater space between particle-bands. Finally, the micro particle image velocity (micro-PIV) experiments were conducted to obtain the bandwidths and spaces between particle-bands. The comparisons between the numerical and experimental results show a good agreement and make the validity of the numerical results certain.

## 1. Introduction

The sorting of microparticles has a great application prospect in the fields of oncology, stem cell research, gene sequencing and so on [1,2]. Inertial microfluidics is often used in microparticle sorting by particle size. The separation efficiency could be improved by using an expansion–contraction microchannel [3,4] or bending microchannel [5,6,7,8] in inertial microfluidics due to the microvortex or Dean vortex. However, the inertial effect is not enough on its own to separate micro-particles with similar sizes. Hence, a variety of approaches to increase the lateral migration of microparticles, by the use of sonophoresis, thermophoresis, dielectrophoresis, photophoresis, etc., were employed in microparticle sorting [9,10,11,12,13].

Magnetophoretic separation is an approach having wide applications in biological medicine and chemical analysis to separate magnetic particles with various magnetic properties or sizes in a viscous fluid [14,15]. Additionally, magnetic microparticles can also be sorted by the shapes of microparticles under a proper applied magnetic field [16,17]. Generally, two approaches to sort the nonmagnetic microparticles are the magnetic beads label [18] and negative magnetophoresis [19,20]. Negative magnetophoresis refers to the movement of nonmagnetic particles away from a magnetic source in a viscous liquid [21]. It is worth noting that the precondition for producing negative magnetophoresis is the magnetic permeability of the medium fluid (i.e., the above-mentioned viscous liquid) being higher than that of the suspending particles. The magnetic buoyancy is related to the magnetic field, the magnetic permeability of the medium fluid and particle, and the particle diameter [22,23,24]. The greatest advantage of the negative magnetophoretic separation of the particle is that there is no need for modification and labeling to the target particles. A magnetic fluid made up of stable and homogeneous dispersed magnetic nanoparticles with a diameter of several nanometers and a carrying liquid normally acts as the medium fluid in negative magnetophoretic separation [25], because the magnetization intensity of the magnetic fluid can be adjusted by the size and volume fraction of the nanoparticles [26].

Increasing research reports of the negative magnetophoretic separation of nonmagnetic particles have been seen in recent years. For example, Xuan’s research group [26,27] presented a ferrofluid-based hybrid microfluidic technique combining passive inertial focusing and active magnetic migration to separate diamagnetic particles by size, and a 3D numerical model to simulate the migration of diamagnetic particles during their inertial focusing and magnetic separation. A simple magnetic technique to concentrate polystyrene particles and live yeast cells in a ferrofluid flow using negative magnetophoresis was demonstrated for various particle sizes, flow velocities and concentrations of MnCl_2_ [28]. Hejazian [29] reported the magnetic manipulation of nonmagnetic particles also suspended in diluted ferrofluid. Various sheath flow ratios, particle sizes and magnetic intensities were used to examine the complex behavior. Fateen [30] and Wang [31] established a three-dimensional FEM model including a magnetic field, flow field and mass transfer equations for the migration of nonmagnetic microparticles induced by negative magnetophoresis. The model successfully predicted different phenomena such as trapping, focusing and deflection. Mao’s research group [22,32] presented an analytical model to predict the particles’ trajectories and the deflections at different flow rates, with different properties of magnetic fluids and different geometrical parameters. A separation device based on negative magnetophoreses was designed, modeled, fabricated and characterized. Yan [33] reported a work to tune and improve the dynamic range of a hydrophoresis device using magnetophoresis. The effects of the flow rate, particle size, magnetic susceptibility of the medium and number of magnets on the particle focusing efficiency were also presented. However, an optimization design for the magnetophoretic sorter has to be conducted to sort 3, 4 and 5 micron-particles because their sizes are too close.

An inertial sorter coupled with a magnetophoretic effect is designed to sort microparticles with similar sizes. The effects of the inlet velocity, sheath flow ratio, particle size, magnetic field and structure parameters of the microchannel on the sorting efficiency will be studied by numerical simulations for various magnetic fields, various geometries and various flows to ensure a group of applicable parameters for sorting microparticles with very similar sizes. Additionally, part of the experiments was carried out to confirm the validity of the numerical results.

## 2. Magnetophoretic Sorter

### 2.1. Negative Magnetophoresis

Magnetophoresis refers to the directional migration of the magnetic particles suspended in a viscous fluid under a magnetic field. The magnetophoretic force exerted on the particle points in the direction of the increasing magnetic field intensity. On the contrary, negative magnetophoresis refers to the movement of nonmagnetic particles away from a magnetic source under a magnetic field [19]. Magnetophoretic force on a particle can be read as [34]:(1)$$F_{m}=\mu_{0}V_{p}\left[\left(M_{p}-M_{f}\right)\cdot \nabla \right]H$$
where $M_{p},M_{f}$ are the magnetization intensity of the particle and medium fluid, respectively. $V_{p}$ is the volume of the particle, H is the magnetic intensity and $\mu_{0}$ is the space permeability. Equation (1) applies to both magnetic and nonmagnetic particles. It is evident that the magnetophoretic force involves the gradient of the magnetic field, the volume of the particle, and the magnetization intensity of the particle and medium fluid. The magnetophoretic force is going in the same direction as the gradient of the magnetic field if the magnetization intensity of the particle is larger than that of the medium fluid (Figure 1a,c–e). It will go in opposite direction if the magnetization intensity of the particle is smaller than that of the medium fluid (Figure 1b). This is right negative magnetophoresis. In addition, it can be concluded from Equation (1) that different particles with different magnetic properties can be separated even if they have the same size. Moreover, different particles with different sizes can also be separated even if they have the same magnetic property. Hence, magnetic fluid with a higher magnetization intensity can act as medium fluid for the separation of nonmagnetic particles by magnetophoresis (Figure 1e).

### 2.2. Force on Microparticles

It is known to us that target microparticles with a certain size should be focused at a certain equilibrium position, so as to be exported at a certain outlet for sorting. Therefore, it is necessary for us to have a good knowledge of the force on microparticles and the movement of microparticles.

The forces on the microparticles in a magnetic fluid involve viscous drag, weight, buoyancy, added mass force, pressure gradient force, Magnus lift force, Saffman lift force, Basset force, Brownian force, thermophoretic force, etc., when no account of the interaction of microparticles suspended in the magnetic fluid is taken (assumed to be a dilute suspension). On the basis of the analysis in Ref. [35], the weight and buoyancy, pressure gradient force and added mass force to the microparticles with a diameter of several microns can be neglected because they are of more than three orders of magnitude less than that of the magnetic force in the high gradient magnetic field. The Magnus and Saffman lift force can also be ignored, because the lift forces are much smaller than the viscous drag. Without taking into account the temperature inhomogeneity and greater flow perturbation, the thermophoretic force and Basset force can also be left out. The stochastic Brownian force is irrelevant, owing to the tiny effect on the directional movement of the microparticles. In short, there are two major factors, the magnetophoretic force and viscous drag, that influence the movement of microparticles with a size of several microns in the magnetic fluid under an applied magnetic field.

The microparticles will be subjected to the resistance of the surrounding fluid due to the viscosity effect when the velocity of the particle is unequal to that of the surrounding fluid. It can be read as:(2)$$F_{D}=-6pmr_{p}\left(u_{p}-u_{f}\right)f_{D}$$
where μ is the dynamic viscosity of the medium fluid, $r_{p}$ is the diameter of the microparticle, and $u_{p}$ and $u_{f}$ are the velocity of the microparticle and medium fluid, respectively. The minus means that the viscous drag is in the opposite direction of the velocity difference between the particle and surrounding fluid. The drag coefficient fD=24/Rep for the present work [36], because the particle Reynolds number is much less than the unit (0.05~0.15).

### 2.3. Geometry of Magnetic Sorter

Based on the above analysis, a new magnetophoresis-coupled inertial sorter for nonmagnetic microparticles is designed (Figure 2). It involves three portions: an inlet region, separation region and enlarged region. The target microparticles are input at inlet A, while the buffer is input at inlet B. Note that 1 vol% of magnetic fluid of Fe_3_O_4_ acts as buffer in the present work. Three contractional sets in the separation region produce two orifices to enhance the lateral migration of the target microparticles due to the microvortex and Dean vortex resulting from the bent streamlines. In the vicinity of the microchannel wall, three magnets are arranged to induce a high gradient magnetic field. The microparticles passing by the magnets will be pushed away from the bottom wall. The setting of the enlarged region serves to enlarge the space between the particle-bands for an easy separation. The dimensions of the microchannel of the present 2D model are shown in Figure 2b. Both inlet A and inlet B are 500 m in width. There are a number of magnets and a microchannel with contraction–expansion geometry in the separation region. The width and length of the contraction channel are H_a_ and L_b_, respectively. The width and length of the expansion channel are H_b_ and (L_a_–L_b_), respectively. H_b_, L_a_ and L_b_ are 500 m, 500 m and 1000 m, respectively.

## 3. Numerical Model

A numerical simulation is a practical approach to investigate the influences of various factors on the sorting efficiency. Hence, a comprehensive numerical model should be established to take into consideration negative magnetophoresis. The numerical model of the present sorting involves a flow calculation, magnetic calculation and particle trajectory calculation.

### 3.1. Flow Calculation

The steady flow calculation in the sorter includes mass and momentum conservation equations if no temperature gradient exists. The mass conservation equation reads as:(3)$$\nabla \cdot u_{f}=0$$

The channel Reynolds number for the present work ranges from 0.5~15. The Knudsen number is right within the scope of the Navier–Stokes equation. It can be read as:(4)$$\rho_{f}\left[\frac{\partial u_{f}}{\partial t}+(u_{f}\cdot \nabla )u_{f}\right]=-\nabla p+\mu \nabla^{2}u_{f}+F$$
where $\rho_{f}$ is the density of the magnetic fluid, p is the pressure and F is the source term which mainly takes into account the magnetic force in Equation (1). It is known from Equation (36) in Ref. [37] that the viscosity of the magnetic fluid varies with the volume fraction of nanoparticles, environment temperature and magnetic intensity. Only the influence of the magnetic intensity on the viscosity needs to be considered because the particle concentration and working temperature in the presented sorter are assumed to be constant. As to the 1 vol% magnetic fluid, the fluid viscosity increases no more than 20% when the external magnetic intensity varies from 0 to 800 Gs [38]. Additionally, the viscosity increases almost linearly with the magnetic intensity to the magnetic fluid with a lower volume fraction.

### 3.2. Magnetic Calculation

It is very important for us to calculate the magnetic field because the magnetic force and magnetic gradient are involved in Equations (1) and (4), respectively. Hence, Maxwell equations are employed to obtain the magnetic flux density within the region of the microchannel [16]:(5)∇×H=0
(6)∇⋅B=0
(7)$$B=\mu_{0}\left(M_{f}+H\right)$$
where B is the magnetic flux density and $M_{f}$ is the magnetization intensity of the magnetic fluid. The influence of the microparticles on the magnetization of the magnetic fluid is not take into consideration because the microparticle is regarded as a dilute phase in simulations.

### 3.3. Microparticle Trajectory

The magnetic force and viscous drag on a microparticle should be taken into account to calculate the particle trajectory, in accordance with the analysis in Section 2.2. Then, Newton’s second law can be written as:(8)$$m_{p}\frac{du_{p}}{dt}=F_{m}+F_{D}$$

The inertia term can be ignored when the velocity variation of the microparticle is not so great. This means that the movement of the microparticle across the streamlines is mainly induced by the magnetic force. The reasonability of such an assumption was verified by Ref. [32]. Moreover, numerical results conducted in advance for lateral migrations are, when ignoring the inertia term, very close to those resulting from considering the inertia term. In addition, the numerical results ignoring the inertia term are more consistent with the subsequent experimental results.

### 3.4. Trial and Verification

The parameter settings (Table 1) for the numerical simulations were carried out, and the commercial software COMSOL for a multiphysical field simulation was employed. After the calculation of the static magnetic field, the dilute suspension flow and particle tracking were coupled to calculate the particle distributions. Meanwhile, the calculation of the width of particle bands (i.e., bandwidth, defined as 99.7% of microparticles being included) and the space between the particle distribution bands (space between PDBs) are processed by programming.

For a simplification of the calculation, a numerical case for the straight microchannel with three magnets nearby was conducted. Grid refining should be carried out near the channel wall to obtain an ideal and smoothly magnetic gradient. The profiles of the magnetic flux density are shown in Figure 3. Both H_x_ and H_y_ in the microchannel, at y = 200 m, vary drastically near the region between magnets (x = 1800~2000 m and x = 2500~2700 m in Figure 4). This means that the magnetic gradients in this region are very great in such a Kietel domain [39].

It is necessary to implement the validation of grid independence before numerical simulations. The discrepancy of the calculated magnetic force of 4 m particles for grid numbers of 64,920, 134,041 and 363,639 was less than 0.068%, when the parameter settings were: the velocity of inlet A and B at 200 m/s and 800 m/s, respectively, and the magnetic field intensity at 0.95 × 10^5^ A/m. Hence, the calculation accuracy was high enough when the grid number was greater than 64,920.

The flow calculation based on Equations (3) and (4) can follow the magnetic calculation because the viscosity and source term in the Navier–Stokes equation are related to the magnetic intensity and magnetization intensity. Similarly, the parameters are set to be the same as above. The numerical results in Figure 5 showed us that the flow velocity of the magnetic fluid in the microchannel was disturbed by the magnets due to the variation of the viscosity and magnetic force.

## 4. Results and Discussions

### 4.1. Effect of Magnets Arrangement

It can be foreseen that the magnetic pole arrangement will affect the magnetic gradient and will subsequently affect the forces on microparticles and the trajectory of microparticles. Two kinds of magnetic pole arrangements named NSN and NNN are simulated for 3 μm, 4 μm and 5 μm nonmagnetic microparticles at the same conditions as mentioned above. Figure 6 indicates that the separation result for the NSN arrangement is much better than that for the NNN arrangement because a greater magnetic gradient in the y-direction can be obtained for the NSN arrangement.

### 4.2. Effect of Magnetic Intensity

It is known from the above analysis that the equilibrium positions of microparticles are mainly determined by the magnetic force and drag force. The magnetic force will affect the sorting efficiency directly and greatly. Hence, numerical simulations for the distributions of various microparticles are carried out under various magnetic fields ranging from 0.5 to 1.1 × 10^5^ A/m when the other parameters are kept constant. Figure 7 shows us that the lateral migrations of microparticles increase almost linearly with the magnetic intensity. Instead, the bandwidths of microparticles show little change with the increase of the magnetic intensity. Furthermore, the magnetic intensity influences the lateral migration of bigger particles (5 μm) more greatly than that of smaller ones. This is why negative magnetophoresis can be employed to separate microparticles by size. It is necessary to arrange bigger magnets to produce a more powerful magnetic field. Nevertheless, the size of a magnet is restricted in a micro-device. One can see that when the magnetic intensity is greater than 0.7 × 10^5^ A/m, three kinds of particles can be distinctly separated, and the space between two PDBs is larger than 50 μm.

### 4.3. Effect of Inlet Velocity

It is well known that the lateral migration of microparticles can be determined by the force acting on the particles and the action time of the magnetic force. The long action time of the magnetic force on the microparticles will result in a greater lateral migration when the external magnetic field is fixed. Numerical simulations for calculating the lateral migration of particles were conducted when the inlet velocity ranged from 1 to 30 mm/s. The computational parameters were set to be: a magnetic intensity of 0.7 × 10^5^ A/m and a sheath flow ratio of 1 (defined as the ratio of the inlet velocity at Inlet A to the inlet velocity at Inlet B). Figure 8 shows us that the lateral migration of the identical particles increases with the decrease of the inlet velocity owing to the longer residence time. Besides, a greater lateral migration can be induced by the bigger particles at an identical inlet velocity. This can be interpreted as the comprehensive results of the magnetic force and drag force, because the magnetic force on the particle is proportional to the cube of the particle size, while the drag force is directly proportional to the particle size.

### 4.4. Effect of Sheath Flow Ratio

The dispersity of particles at the entrance can be controlled by the sheath flow [40], i.e., the PDBs can be compacted by the sheath flow. Sheath flow can also be employed in the magnetophoretic separation process to improve the separation efficiency. Numerical simulations for various sheath flow rates (1–10) are conducted to obtain particle distributions (Figure 9) at a constant velocity of 1.2 mm/s in the main channel (i.e., the sum of the velocity at inlet A and inlet B). As can be seen from the figure, the bandwidths of the three kinds of particles gradually decrease with the increase of the sheath flow ratio. What is more noteworthy is the fact that the space between PDBs also increases with the increase of the sheath flow ratio, which is conducive to particle separation. The reason for this is that the PDBs can be contracted closer to the bottom wall near the magnets by the sheath flow, and a greater magnetophoretic force can be exerted on the particles due to the greater magnetic gradient (Figure 6) when the sheath flow ratio is greater. Furthermore, Figure 9 also indicates that the PDBs for different particles cannot be separated when the sheath flow ratio is less than 2, and that the spaces between PDBs are hardly changed when the sheath flow ratio is greater than 7. After comprehensive consideration, it can be concluded that the optional scope of sheath flow ratios is 4–7 if the required space between PDBs can be expected. However, the preferential value is 4 for a high production rate because the sample inlet velocity is the maximum when the sum of the sample and buffer inlet velocity is constant.

### 4.5. Effect of Geometrical Parameters

To further improve the sorting efficiency, a channel with a contraction–expansion geometry can be employed to act as a separation channel because of the inertial effect induced by the microvortex in the expansion region and the Dean vortex. The bandwidths and the space between PDBs listed in Table 2 are calculated through a series of simulations for various contraction–expansion ratios (H_a_/H_b_ = 0.6, 0.5 and 0.4) and external magnetic intensities (H = 0.5–0.7 × 10^5^ A/m). The other parameters are set to be: L_a_ = 500 μm, L_b_ = 1000 μm, H_b_ = 500 μm, u_A_ + u_B_ = 1.2 mm/s and sheath flow ratio = 4. It can be seen from Table 2 that the magnetic intensity will obviously affect the space of PDBs, and will instead lightly affect the bandwidths.

In addition, the bandwidths and the space between PDBs listed in Table 3 for H = 0.7 × 10^5^ A/m indicate that the spaces between two PDBs increase with the decrease of the contraction–expansion ratio. There are two reasons for this: the first reason is a greater magnetic force F_y_ caused by the squeeze of PDBs closer to the bottom wall near the magnets; the other reason is a greater lateral migration for larger particles because of the movement across the streamlines induced by the inertial effect. The smaller the contract–expansion ratio is, the better choice there is for every case if the pressure drop is not taken into account. However, H_a_/H_b_ = 0.5 is preferential if both factors are taken in consideration. Moreover, there is no need to worry about the trapping of larger particles in expansion regions (i.e., orifices) because the distances between PDBs and the upper wall are great enough due to the sheath flow.

## 5. Experimental Results

### 5.1. Experimental Material

#### 5.1.1. Preparation of Nonmagnetic Microparticle

First, Fe_3_O_4_ magnetic fluids including nanoparticles with average sizes of 9.8 nm, 14.6 nm and 21.2nm (Titan™ G2 60–300) were prepared by sol-gel method. Then, a polystyrene microsphere was synthesized by the emulsion polymerization method, that is: the dispersive polymerization of styrene, acrylic acid and divinyl benzene in the existing magnetic fluid using the dispersion of polyethylene glycol and the dispersive medium of ethyl alcohol/water. Three kinds of microparticles with average sizes of 2950 nm, 3875 nm and 5085 nm (Microtrac S3500) and magnetization intensities of 2 Gs, 12 Gs and 22 Gs (VSM-350) were obtained. The microparticles can be regarded as being nonmagnetic owing to their lower magnetization intensity.

#### 5.1.2. Sorter Manufacture

Sorters with contraction–expansion channels for separating nonmagnetic microparticles were manufactured (Figure 10) in light of the analysis in 1.2. The channel widths and depths for inlets and the main channel are 500 m and 300 m, respectively. For comparison, two sorters with the contraction–expansion ratios H_a_/H_b_ = 1 and 0.5 were prepared (H_a_/H_b_ = 1 representing the straight channel). Three NdFeB (neodymium iron boron) magnets spaced with two copper pieces were placed near the channel, and the magnetic intensities were regulated and controlled by the length of magnets.

### 5.2. Experimental Setup

An experimental setup (Figure 11) whose core was Dantec Micro-PIV was composed of four modules: laser system, amplifier and camera, image post-processing and microfluidic system [40]. The laser system was composed of a dual-wavelength laser, power source and synchronizer. The parameters for ND: YAG laser (LAS036): maximum energy: 400 mJ, pulse interval: 4 ns, wave-length: 532 nm. The amplifier was a microscopy (Dentec 80M57) with a 20 × magnification of objective lens, and the parameters for high-speed CCD (VSC-04253): pixel: 2048 × 1700, frequency: 280 Mhz, interval: 100 s. Image post-processing was system-provided via Dynamic studio 2015a. The microfluidic system included an integrated micropump, a number of pipelines and a sorting chip. The micropump (WH-MPMM-15) was integrated by a 15-channel constant pressure pump, microvalves and a pressure controller.

### 5.3. Experimental Results

To intuitively understand the effect of the magnetic field and geometry on the particle movement and separation process, numerical and experimental investigations for 3 μm, 4 μm and 5 μm particles were conducted. The corresponding parameters were: u_A_ = 800 μm/s, u_B_ = 200 μm/s, u_A_/u_B_ = 4, H = 0.58 × 10^5^ A/m, and H_a_/H_b_ = 0.5 and 1. A series of negative magnetophoretic sorting experiments for various magnetic intensities (0.5 and 0.7 × 10^5^ A/m), sheath flow ratios (1 and 4) and H_a_/H_b_ (1 and 0.5) were carried out. Figure 12 qualitatively shows good agreements between the numerical results and experimental results (Figure 12). The quantitative results are listed in Table 4, the errors being less than 5%. It can be seen that the sorting efficiency increases with the magnetic intensity and sheath flow ratio, while decreasing with the contraction–expansion ratio instead. Note that the bandwidth and space between PDBs cannot be read out from the experimental images when u_B_/u_A_ = 1 because of the overlap of PDBs (see in Figure 12d); hence, only numerical results are listed in Table 4.

## 6. Conclusions

A new negative magnetophoretic sorter with a contraction–expansion channel for microparticles with a very close diameter was designed. The numerical model, involving a magnetic model, fluid model and particle model, was established in allusion to three kinds of microparticles with diameters of 3, 4 and 5 μm. The particle trajectories and corresponding particle distributions were calculated for various particles under various conditions. The following conclusions can be drawn:(1)The lateral migrations for three kinds of microparticles increase with the magnetic intensities and particle sizes. The preferential value of the magnetic intensity is ~0.7 × 10^5^ A/m, because the size of magnets is limited in a microfluidic system. In addition, the NSN arrangement can result in a satisfied sorting efficiency.(2)The sorting efficiency rises when the inlet velocity decreases. However, an overly low inlet velocity for the sample results in a lower productivity. The sorting efficiency increases with the sheath flow ratio. The satisfied bandwidths and spaces between PDBs can be produced with a sheath flow ratio ranging from 4 to 7. When taking into account the productivity, the preferential value of the sheath flow ratio is 4.(3)The spaces between PDBs increase with the decreasing of the contraction–expansion ratio (H_a_/H_b_) because of the combination of the negative magnetophretic effect and inertial effect. The repeated compression of PDBs close to the magnets can be induced by the repeated contraction of the channel. When taking into account the pressure drop, the preferential value of the contraction–expansion ratio is 0.5.

## Figures and Tables

**Figure 1 micromachines-11-00566-f001:**
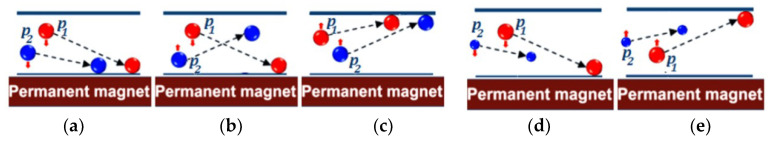
Schematic of particle separation by magnetophoresis. (**a**) $M_{p1}\geq M_{p2}\geq M_{f},V_{p1}=V_{p2}$, (**b**) $M_{p1}\geq M_{f}\geq M_{p2},V_{p1}=V_{p2}$, (**c**) $M_{f}\geq M_{p2}\geq M_{p1},V_{p1}=V_{p2}$, (**d**) $M_{p1}=M_{p2}\leq M_{f},V_{p1}\geq V_{p2}$, (**e**) $M_{p1}=M_{p2}\leq M_{f},V_{p1}\geq V_{p2}$.

**Figure 2 micromachines-11-00566-f002:**
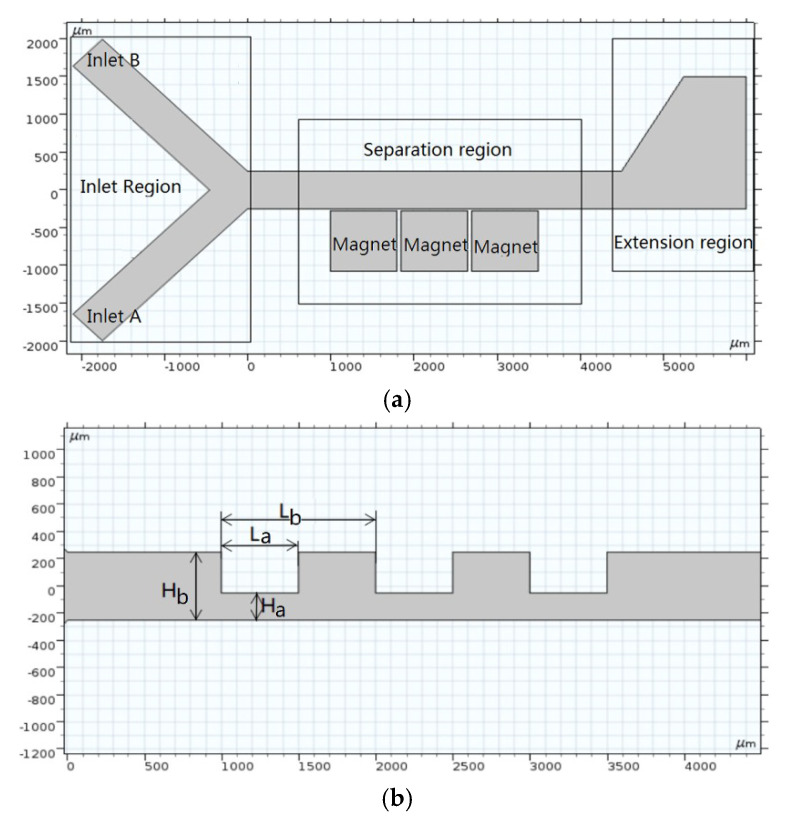
(**a**) Schematic diagram and (**b**) geometrical dimensions of the negative magnetophoretic sorter.

**Figure 3 micromachines-11-00566-f003:**
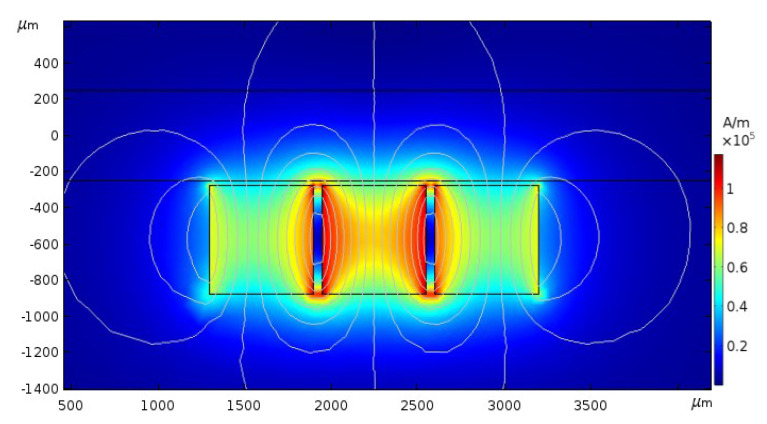
Profiles of the magnetic flux density.

**Figure 4 micromachines-11-00566-f004:**
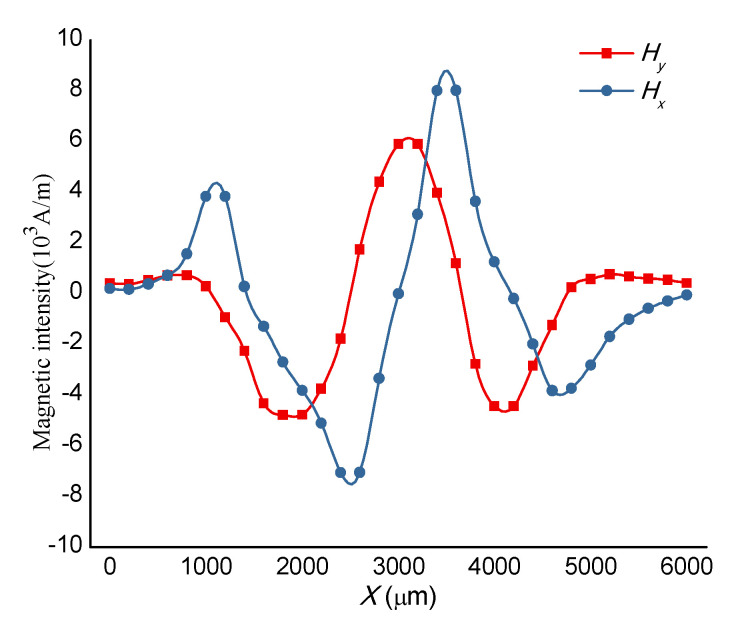
Distribution of the magnetic intensity at y = 200 m.

**Figure 5 micromachines-11-00566-f005:**
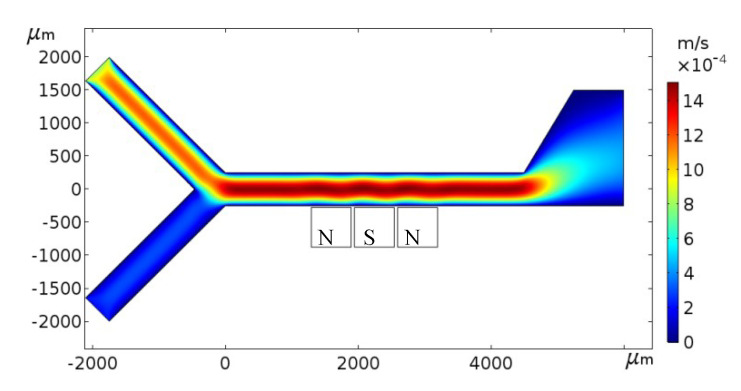
Velocity profile near the magnets.

**Figure 6 micromachines-11-00566-f006:**
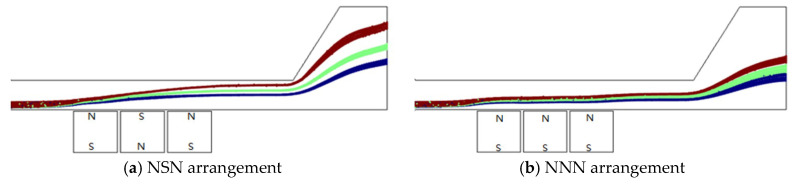
Particle trajectories for various magnetic pole distributions. (**a**) NSN, (**b**) NNN.

**Figure 7 micromachines-11-00566-f007:**
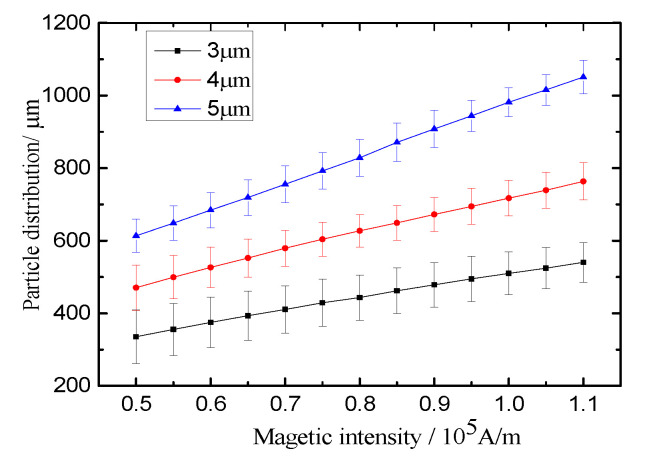
Effect of magnetic fields on the particles’ distributions.

**Figure 8 micromachines-11-00566-f008:**
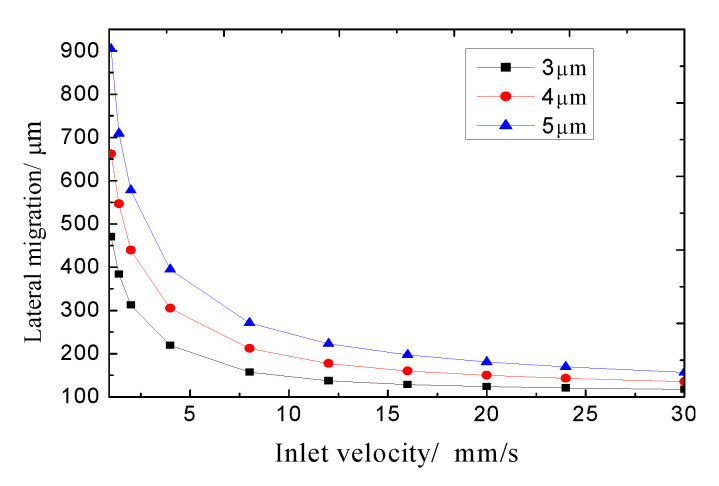
Lateral migrations of micro-particles with various inlet velocities.

**Figure 9 micromachines-11-00566-f009:**
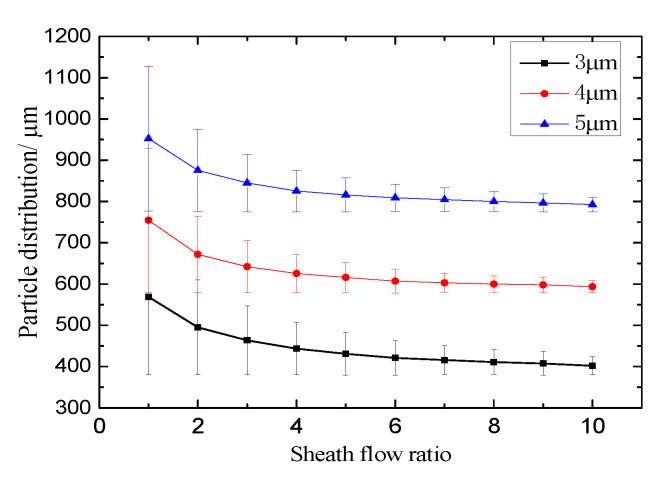
Distribution of various particles at various sheath flow ratios (H = 0.7 × 10^5^ A/m, u_A_ + u_B_ = 1.2 mm/s).

**Figure 10 micromachines-11-00566-f010:**
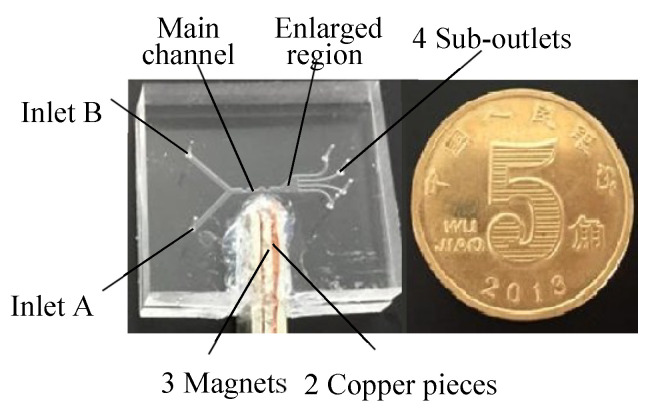
Sorter with the contraction–expansion channel for negative magnetophoresis.

**Figure 11 micromachines-11-00566-f011:**
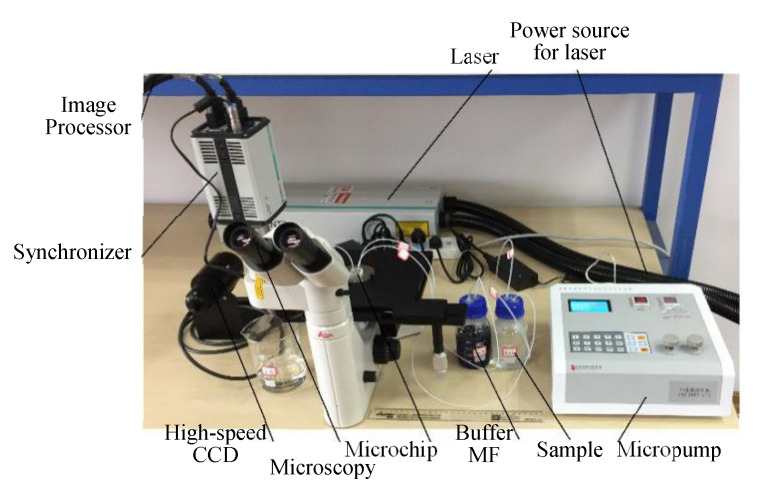
Experimental setup for laser-induced fluorescence.

**Figure 12 micromachines-11-00566-f012:**
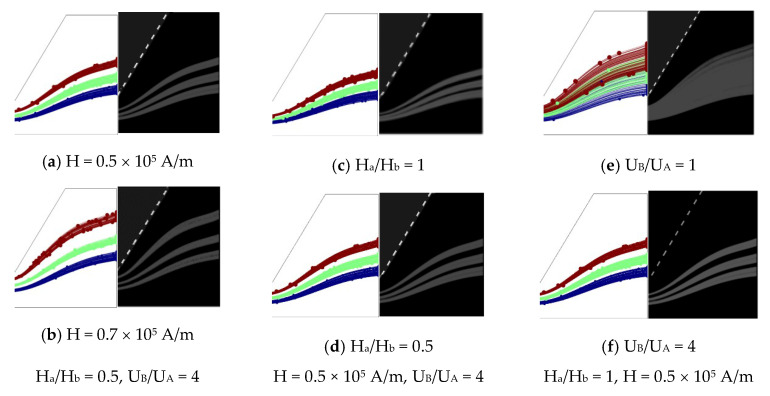
Qualitative comparison of the numerical and experimental results.

**Table 1 micromachines-11-00566-t001:** Parameters for the numerical simulation.

Notation	Value	Description
*µ_0_*	4π × 10^−7^ H/m	Space permeability
*η_f_*	1.029 mPa.s	Viscosity of MF
	1%	Volume fraction of nanoparticles in MF
*M_f_*	2.2 × 10^5^ A/m	Saturation magnetization of MF
*_f_*	1133 kg/m^3^	Density of MF
*_p_*	1058 kg/m^3^	Density of microparticles
*M_s_*	1.5 × 10^5^ A/m	Remanent magnetization of magnets
*M_p_*	0	Magnetic susceptibility of microparticles

**Table 2 micromachines-11-00566-t002:** Bandwidths and spaces between PDBs.

H_a_/H_b_ = 0.4					
Magnetic Intensity	Bandwidth 3 μm	Bandwidth 4 μm	Bandwidth 5 μm	Space between PDBs 3–4 μm	Space between PDBs 4–5 μm
0.50	157	123	156	55	104
0.54	160	123	151	74	144
0.58	159	128	147	83	177
0.62	151	132	168	85	194
0.66	149	137	233	111	228
0.70	144	134	238	127	278
0.74	139	147	283	155	374
H_a_/H_b_ = 0.5					
0.50	145	138	135	49	92
0.54	147	138	136	53	106
0.58	147	134	149	68	126
0.62	148	129	158	83	142
0.66	148	127	160	96	176
0.70	148	124	160	108	205
0.74	148	119	188	121	246
H_a_/H_b_ = 0.6					
0.50	150	129	121	33	67
0.54	150	126	124	47	80
0.58	149	120	128	60	99
0.62	146	115	134	74	120
0.66	142	114	134	86	142
0.70	140	114	131	96	167
0.74	135	117	126	105	190

**Table 3 micromachines-11-00566-t003:** Bandwidths and spaces between PDBs.

H_a_/H_b_	Bandwidth 3 μm	Bandwidth 4 μm	Bandwidth 5 μm	Space of PDBs 3–4 μm	Space of PDBs 4–5 μm
1	131	99	101	54	76
0.6	140	114	131	96	167
0.5	148	124	160	108	205
0.4	144	134	238	127	278

**Table 4 micromachines-11-00566-t004:** Quantitative comparison of the numerical and experimental results.

*H* = 0.5 × 10^5^ A/m, H_a_/H_b_ = 0.5, U_B_/U_A_ = 4, U_A_ + U_B_ = 1000 μm/s					
Index Name	Bandwidth 3 μm	Bandwidth 4 μm	Bandwidth 5 μm	Space of PDBs 3–4 μm	Space of PDBs 4–5 μm
Numerical	145	138	135	49	92
Experimental	148	142	136	47	95
Error %	2.07	2.90	0.74	4.08	3.26
*H* = 0.7 × 10^5^ A/m, H_a_/H_b_ = 0.5, U_B_/U_A_ = 4, U_A_ + U_B_ = 1000 μm/s					
Numerical	148	124	160	108	205
Experimental	147	128	157	111	201
Error %	0.68	3.22	1.88	2.78	1.95
*H* = 0.5 × 10^5^ A/m, H_a_/H_b_ = 1, U_B_/U_A_ = 4, U_A_ + U_B_ = 1000 μm/s					
Numerical	122	116	114	36	71
Experimental	125	118	113	37	68
Error %	1.64	1.72	0.88	2.78	4.17
*H* = 0.5 × 10^5^ A/m, H_a_/H_b_ = 0.5, U_B_/U_A_ = 1, U_A_ + U_B_ = 1000 μm/s					
Numerical	328	294	271	−58	−71

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
