# Peer review of "Investigation on Inertial Sorter Coupled with Magnetophoretic Effect for Nonmagnetic Microparticles"

_micromachines, 2020, doi:10.3390/mi11060566_

Round 1
Reviewer 1 Report
I wrote all comments in the PDF.

Author Response
To reviewer 1
Thanks for your comments. We get a lot of inspiration from the comments.
Summary:
The authors investigated the migration of small non-magnetic particles based on the negative-magnetophoretic effects, both numerically and experimentally. They used a software COMSOL for their simulations, and the fluid mechanics and the Maxwell’s equation are couple to compute the particle trajectories. Due to the magnets that are placed close to the microfluidic channel, the particles migrate away from the wall. Since the particle position depends on the particle sizes, the particles with different radii can be separated. The numerical results have good agreement with the experimental result, though it seems that the numerical conditions have some many discrepancies (as pointed out in the major comment) from the experimental conditions.
The manuscript was easy to follow, but the quality of scientific writing and presentation was bad. There are so many typos and mistakes in the document (for example, they even made a typo in their title, “Magnetophretic”), and the authors should pay intense effort to improve their manuscript quality. The manuscript is losing the trust because of these bad science writing (as I listed in the minor comments).
I do not think the result is not well established and written to be published. A drastic editing is necessary, if the paper is going to a next round.
Major Comments:
- Although the authors included in the title words “prokaryotic cells”, they did not used these cells in their experiments. Example of the future application should not be included in the title. I strongly recommend removing the words. Or, at least change to more general words.
Response: Okay, we remove “prokaryotic cells”
- Introduction: Please write down explicitly what is the novelty of this work. There are already theories for the negative-magnetophoretic migration as pointed out in the introduction. What is the novelty of this work? (sheath flow ratios? contraction-expansion ratio? particle sorting? or allthese?)
Response: The microparticles with a size of 3,4,5 micron can be separated and sorted only at appropriate parameters obtained from our works(including magnetic intensity, sheath flow ratio, contraction-expansion ratio, magnets arrangement, inlet velocity) . The purposes of present work are to obtain a group of preferential parameters for successful separation of microparticles with very similar sizes, and to explore the law of influence of applied magnetic field, flow condition and geometrical parameters on sorting efficiency.
- There are some interesting works that control the particle migration using ellipsoidal particles in recent years (https://doi.org/10.1039/C6LC01382A, https://10.1103/PhysRevLett.119.198002). I recommend citing these
Response: Those are interesting and great works, we cited them.
- What is the Reynolds number (and/or the particle Reynolds number) of this system, both in simulation and experiment? Please write down in the
Response: We give the channel Re and particle Re for present work in revised manuscript.
- I cannot understand the following sentences: “A simplification to ignore the inertia term can be introduced when the velocity fluctuation of magnetic fluid and velocity variation of microparticle being not so violent. … The reasonability of such a assumption can be verified bytrials”.
5a. Does it mean the authors ignored the inertia term? If yes, it can be ignored if the particle Reynold number is far small compared to unity. “velocity fluctuation ... being not so violent” is not a correct criterion.
Response:Yes, the inertia term is ignored in present work(Rep=0.05-0.15) . The word “violent” is replaced by “great”.
5b. What do you mean by trials? Please explain clearly what kind of “trials” the authors did in this work.
Response: We conducted some trials to calculate the lateral migrations of microparticles at inlet velocity ranging 1-30mm/s by ignoring and considering inertia term, and found the differences do not exceed 3%. Moreover, the results ignoring inertia term is more consistent with the experimental results than that considering inertia term. Some interpretations are added.
- The authors did not mention whether the COMSOL simulation is in 2D or 3D (I assume it was 2D simulation because there is no channel height information). Please write down clearly. At the same time: if the simulation is in 2D, what is the meaning of considering the density difference between MF and microparticle? and where is the gravitational body force in Eq. (8)?
Response: 2D. The purpose to consider the density difference between MF and particles are: 1) weight+buoyancy~0 and can be neglected, 2) It corresponds to the density of the most prokaryotic cell. Some interpretations are added.
- Although it is said “the viscosity and source term in Navier-Stockes equation are related to magnetic intensity and magnetization intensity”, there is no explicit description how the viscosity and source term is the function of the magnetic intensity H. Please write down explicitly.
Response:Coupling computations for flow field and particle trajectory are carried out after magnetic calculation. The viscosity and source term can be calculated by Eqn 36 in Ref[37] and Eqn 1 in present manuscript, respectively. Some interpretations are added.
- I thought NNN arrangement should push the particle further compared to NSN, because NNN can push the particle continuously in the same direction. Please explain in the manuscript why NSN works better. (or, explain with a new figure with a clear evidence “greater magnetic gradient in y-direction can be obtained for NSNarrangement”.)
Response: Note, we concerned nonmagnetic particle. It is known from Eqn.1, the magnetic gradient is related to the magnetic force Fm on a nonmagnetic microparticle. The below figures show that, there are greater influence on the magnetic field in main channel for NSN than that for NNN, especially in the region between 2 magnets (seen in Figure below).
- Figure 7: I cannot understand why the particles show the maximum migration at intermediate number of magnets, 3. Why the curve is not converging to a certain value (migration with 3 magnets and 4 magnets is the same), and why it goes down for 4 magnets? The explanation, “the PDBs were pushed away from the region near the bottom wall where great magnetic gradient exists” is not appropriate because it does not explain why the migration decreases for 4
Response: Thanks. It is more acceptable if the increment of lateral migration decreases substantially with the number of magnets, because the magnetophoretic forces on a particle decrease obviously when the particle being far away from bottom wall. Below figure show us the magnetophoretic forces on a particle at y=0 (at the center of channel) and y=200(near upper wall) is about 1/10 and 1/100 of that at y=-200(near bottom wall).
However, the fourth magnet should be arranged near the enlarged region, the upwards y-direction velocity of fluid induce reverse viscous drag which can result in the microparticles to migrate in reverse across the streamlines. This influence on 5μm microparticle is more remarkable than that on 3μm microparticle because bigger microparticle will be driven further away and greater reverse viscous drag.
- Page 8, line 209: What is the explicit definition of the particle “bandwidth”? Is it the maximum and minimum of the distribution, or the standard deviation of the distribution? Please write down
Response: Yes, in 3-s(standard deviation of the distribution). The bandwidth calculated from the concentration distribution involves 99.7% microparticles. Some interpretations are added.
- Page 10, line 244: Explain why “the preferential value is 4 for high productionrate”.
Response:The optional scope of sheath flow ratios is 4-7, the inlet velocity of sample (uA)is maximum (i.e. maximum production rate) when the minimum value being 4, because the sum of sample inlet velocity and buffer inlet velocityare constant(uA+uB=Const) . Some interpretations are added.
- Page 11, line 261: I could not understand this sentence: “it needn’t … the bottom wall”. I thought that, the more the particle are pushed away from the bottom wall, there are higher chance to be trapped to the orifices or the recirculation regions. Please explain the
Response: Yes, you are right. The statement in manuscript is improper. We revised it.
- Tables: I have no idea why the authors suddenly started to display the results in table. No reader wants to check the results in detail line-by-line by numbers. Please change the tables into figures as many as possible. If there is an appropriate reason using the tables, please explain.
Response: It is easy to separate and sort the particles when the spaces of PDBs is great enough(normally, >50mm are expected). Hence we listed in Table 2 both bandwidths of various particles and the spaces of PDBs between 3-4mm and 4-5mm. We did try to display these results in figures(See below figures). Unfortunately, we didn’t satisfy with them, because we can not compare the bandwidths. Instead, we can produce 3 figures like Figure 8 or 10 to show both the bandwidths and space between PDBs. However, it is difficult for us to compare the results at various Ha/Hb because the results are depicted in their own figures. In our opinion, table maybe a better option.
Figure 11 Space between two PDBs(3mm and 4mm, 4mm and 5mm )at various contraction-expansion ratios (Ha/Hb=0.4,0.5,0.6) when H=0.5-0.72×105 A/m, uA+uB=1.2mm/s.
Figure 11 Space between two PDBs(3mm and 4mm, 4mm and 5mm )at various contraction-expansion ratios (Ha/Hb=0.4,0.5,0.6) when H=0.5-0.72×105 A/m, uA+uB=1.2mm/s.
Minor Comments:
Thanks for reviewer’s minor comments. We feel very sorry because we made so many mistakes and typos. We checked the whole manuscript carefully and revised according to reviewer’s comment.
As I pointed out in the summary, the quality of scientific writing (apart from the result itself) was really bad. The authors should pay more attentions to their manuscript quality.
- Title: typo“Magnetophretic”
Response: Revised.
- Page 1, line 36: “respectively” is used in a wrong way. “A and B are C and D, respectively” is the right
Response: Revised.
- Page 2, line 49 and 60: Expressions “Group Xuan” and “Group Mao” are wrong. change to “Xuan/Mao’s research group” or
Response: Revised.
- Page 4, line 120: typo“microparticlesdue”
Response: Revised.
- Page 5, line 138 and Page 6, line 181: typo“Navier-Stockes”
Response: Revised.
- Page 5, line 159: I never heard an expression “velocity variation …being not so violent”. Use other proper word, and do not use
Response: Revised.
- Page 6, line 176: Are you sure that inlet A is 800 um/s and B is 200 um/s? I thought that the value is other way around from the
Response: Yes, you are right. We made a mistake. Revised.
- Page 6, line 182: What is “in 2.4.1”? There is no such figure or
Response: Revised.
- Page 7, Figure 5: The letters “N-S-N” are not placed at the square
Response: Revised.
- Page 8, line 197: again, “in4.1”.
Response: Revised.
- Page 9-10, Figures 7-10: Unify the colors for the particle size. 4um is red for figures 7 and 9, but 5um is red for figures 8 and 10. They are super
Response: Revised.
- Page 10, line 233: “to obtain theto particle distribution”
Response: Revised.
- Page 10, line 233: “Figure 13” is wrong. It is “Figure10”
Response: Revised.
- Page 10, line 236: “the spacebetween PDBs”
Response: Revised.
- Page 12, line 282: typo“microphere”
Response: Revised.
- Page 13, Figure 11: Explanation “Main channel” is behind the
Response: Revised.
- Page 14, Table 4: Some of the numbers in a column “Bandwidth 4um” is aligned left, though all other numbers are aligned at the
Response: Revised.
- Page 15, line 327: What do you mean by “mortal”? I cannot understand this sentence because of this word. I also strongly believe that it is a bad idea to use a word in the conclusion that was never used before in the
Response: Revised.
- Page 15, line 329: change “too lower” to “toolow”
Response: Revised.

Reviewer 2 Report
The paper idea may be sound but unfortunately the experimental part is not sufficiently developed. In my opinion the data provided should be sustained by a number of increased experiments and results interpretations should contain more details!
The paper may be published case the authors succeed in improving the results interpretation!
Author Response
To reviewer 2
Comments and Suggestions for Authors
- The paper idea may be sound but unfortunately the experimental part is not sufficiently developed. In my opinion the data provided should be sustained by a number of increased experiments and results interpretations should contain more details!
Response: Thanks for your suggestion. Limited to the experimental conditions, only part of the experiments were carried out to verify the numerical results. The main purpose of present paper is to ascertain the relevant parameters of sorter(such as magnetic parameters, flow parameters and geometric parameters) by means of numerical simulations, in order that the particles with very similar sizes can be separated and sorted. As is beneficial to design a sorter for non-magnetic microparticles with very similar sizes. Some interpretations are added.
- The paper may be published case the authors succeed in improving the results interpretation!
Response: It is true that there are some unclear points in the manuscript. We have carefully examined and revised. Thank you.
Round 2
Reviewer 1 Report
Thank you for reflecting my suggestions.
I would like to accept after the following points are revised.
- Two of my comments were not reflected in the manuscript, and should be revised. [Minor comment 8] The letters "N-S-N" are still not at the square center. Instead, there are weird "N-S-N" letters on page 7, line 186. [Minor comment 15] The words in Figure 11 ("Main ... ", maybe main channel) are still behind the photo.
- [Major comment 9] I read your statement, but I believe this statement is wrong. If the particle is in the enlarged region, the particle goes up together with the flow as also shown in your experiments, Figure 13. I never heard of "reverse viscous drag" and I do not see any force that tries to pull the particle down. It is totally up to the authors: but I recommend to remove the explanation, or add more explanation if the authors think the statement is correct.
Author Response
Response to Reviewer 1
Thank you for your comments.
I would like to accept after the following points are revised.
- Two of my comments were not reflected in the manuscript, and should be revised. [Minor comment 8] The letters "N-S-N" are still not at the square center. Instead, there are weird "N-S-N" letters on page 7, line 186. [Minor comment 15] The words in Figure 11 ("Main ... ", maybe main channel) are still behind the photo.
Response:I revised them. The cause may be, I use WPS, the reviewer use Words.
I will send corresponding PDF document.
- [Major comment 9] I read your statement, but I believe this statement is wrong. If the particle is in the enlarged region, the particle goes up together with the flow as also shown in your experiments, Figure 13. I never heard of "reverse viscous drag" and I do not see any force that tries to pull the particle down. It is totally up to the authors: but I recommend to remove the explanation, or add more explanation if the
authors think the statement is correct.
Response:We decide to delete “4.2 Effect of magnets number”, because there is no effect on the main idea of the paper, I think. In addition, there are no relevant representations in “ABSTRACT” and in “CONCLUSIONS”.
Further Explanations:
- Zhuo QY finished some primary research works including “Effect of magnets arrangement”and “Effect of magnets number”. He is now out of school for graduation. His main findings include: 1) NSN being better than NNN, 2) the magnets position should not be too close to the junction of sample and buffer, nor too close to enlarged region. The lateral migration will be too large to touch the upper wall when the position is too close to the junction(inlet region). However, the lateral will be too small when the magnets position is too close to enlarger region, this is not conducive to particle separation. 3) 3 magnets being preferential.
- We cannot find suitable interpretations for this point really. What Zhuo QY interpreted is the downwards viscous drag can be induced when the fluids have upwards velocity due to the inertia of particle. This means the particles near enlarged region can cross the streamlines in reverse direction comparing to the particles subjected magnetophoretic force in upstream region.